# A Study on 3D Deep Learning-Based Automatic Diagnosis of Nasal Fractures

**DOI:** 10.3390/s22020506

**Published:** 2022-01-10

**Authors:** Yu Jin Seol, Young Jae Kim, Yoon Sang Kim, Young Woo Cheon, Kwang Gi Kim

**Affiliations:** 1Department of Biomedical Engineering, Gachon University, 191, Hambangmoe-ro, Yeonsu-gu, Incheon 21936, Korea; tjfwlgns0518@gmail.com; 2Department of Biomedical Engineering, Gachon University College of Medicine, 38-13 Docjeom-ro 3 beon-gil, Namdong-gu, Incheon 21565, Korea; youngjae@gachon.ac.kr; 3Department of Plastic and Reconstructive Surgery, Gachon University Gil Medical Center, College of Medicine, Incheon 21565, Korea; yunsang0115@gmail.com; 4Department of Health Sciences and Technology, Gachon Advanced Institute for Health Sciences and Technology (GAIHST), Gachon University, Seongnam-si 13120, Korea

**Keywords:** artificial intelligence, computed aided diagnosis (CAD), 3D-classification, nasal fractures

## Abstract

This paper reported a study on the 3-dimensional deep-learning-based automatic diagnosis of nasal fractures. (1) Background: The nasal bone is the most protuberant feature of the face; therefore, it is highly vulnerable to facial trauma and its fractures are known as the most common facial fractures worldwide. In addition, its adhesion causes rapid deformation, so a clear diagnosis is needed early after fracture onset. (2) Methods: The collected computed tomography images were reconstructed to isotropic voxel data including the whole region of the nasal bone, which are represented in a fixed cubic volume. The configured 3-dimensional input data were then automatically classified by the deep learning of residual neural networks (3D-ResNet34 and ResNet50) with the spatial context information using a single network, whose performance was evaluated by 5-fold cross-validation. (3) Results: The classification of nasal fractures with simple 3D-ResNet34 and ResNet50 networks achieved areas under the receiver operating characteristic curve of 94.5% and 93.4% for binary classification, respectively, both indicating unprecedented high performance in the task. (4) Conclusions: In this paper, it is presented the possibility of automatic nasal bone fracture diagnosis using a 3-dimensional Resnet-based single classification network and it will improve the diagnostic environment with future research.

## 1. Introduction

The nasal bone is the most prominent part of the facial skeleton, making it more vulnerable to traumatic fractures. Therefore, nasal fractures are the most frequent facial fractures worldwide and can be caused by relatively weak forces [1,2]. Furthermore, nasal fractures due to facial trauma retain their state in a relatively short time, within 1–2 weeks, causing nasal deformity with a high incidence of approximately 14 to 50 percent [3]. Rhinoplasty for nasal deformity from a facial trauma is one of the most challenging problems for surgeons; thus, nasal fractures require accurate diagnosis immediately after the onset and before deformation occurs [4]. To perform sophisticated diagnosis of nasal fractures and determination of the range and pattern of a fractured nose, radiologists have recently used computed tomography (CT) images, mostly achieving excellent sensitivity and specificity [5].

The diagnosis of traumatic nasal fractures from facial CT analysis is an effective tool for their early reconstruction. However, the process of CT image reading not only burdens readers (radiologists) with the need to examine numerous CT images in great detail but also makes it difficult to detect fine nasal bone defects in fractures, with high subjectivity in the diagnostic results reported by each reader [6]. Thus, the diagnosis of nasal fractures based on facial CT images requires time-consuming and repetitive work by the reader. Multiple readers are required to verify the images discontinuously in duplicate to increase objective accuracy [7]. In the diagnosis of nasal fractures that require rapid and accurate determination to prevent nasal deformity and further complications, computer-aided diagnostic systems (CAD) can be useful, as they have been reported to have high performance in the automatic diagnosis of various types of fracture [8,9,10]. CAD systems can improve CT image-based diagnosis of nasal fractures by using deep learning algorithms able to automatically discriminate between fractured nasal bones and normal bones [11].

Our task was to develop a binary classification system to detect nasal fractures from facial CT scans using a CAD system based on a 3-dimensional-convolutional neural network (3D-CNN), which analyzes the input images in three dimensions, thus making the 3D (3-dimensional) context information of the input image data far more available than in 2D-CNNs [12]. In addition, a CAD system using deep learning with 3D-CNN can determine the structure and fracture of the three-dimensional bone, thus increasing diagnostic accuracy [13]. In the last decades, several studies have reported that 3D-CNNs can perform labeling of samples from 3D rendering data by deep learning [14,15]. Since their early stages of development 3D-CNNs have attracted attention for their use in the automatically aided diagnosis, and high discrimination performance has been reported when the diagnostic process needs to assess overall anatomic structures [16,17]. Automatic classification by a 3D-CNN typically includes a segmentation step for the rendering of objects. These results in multiple networks and heavy learning models. In such cases, applying limited medical data to heavy learning models can lead to overfitting. Therefore, this study used volume of interest (VOI) input data obtained from image pre-processing to include only the nasal bone region of interest from facial CT images before learning, excluding segmentation steps. The input data is processed, including the nasal bone regions of each patient, into cubic voxel data derived from 3D spatial information from each CT scan. This allows the use of a single classification network and reduces the possibility of overfitting.

In this study, the experiment was performed as a binary nasal bone classification to discriminate between normal and fractured based on a 3D-CNN applied to the collected nasal CT images. And then, it applied 3D-CNN architectures for learning and extracted not only distinguishable features from nasal images, but also, more effectively than with a 2D-CNN, spatial information that is encoded across the nasal CT scans. In addition, this study demonstrates that 3D-CNN with a classification network can result in high performance in identifying fractures of the nasal bone when using limited medical facial data without an extra rendering process. It means that it results in significant CAD performance with a single learning process compared to the already published works. It is faster than passing through multiple networks. To the best of our knowledge, this is the first work that classifies the normal nasal region and fractured nasal region using a single CNN only.

## 2. Materials and Methods

### 2.1. Ethics Statement

This study was approved by the International Review Board of Gil Medical Center (GBIRB2020-091).

It was conducted within a research project of the Ministry of Science and ICT and the Institute of Information about Communication Planning and Evaluation to foster the University of ICT Research Center. Informed consent was obtained from all patients at this institution. All experiments in this study were performed with the relevant guidelines and regulations in accordance with the Declaration of Helsinki.

### 2.2. Data

In this study, facial CT images were obtained from a total of 2535 patients who underwent facial CT at Gil Medical Center, including 1350 normal nasal bones and 1185 fractured nasal bones. To exclude the effects of bone deformation after fracture, only cases were used in which CT was performed less than two days after fracture onset (i.e., the mean time after fracture before the scan = 1.49 days). The examples of 3D rendering reconstruction of normal and fractured nasal bones showing the anatomically structural features in each class are shown in Figure 1a,b. Figure 1 is created through reconstruction with volume rendering of a 3D slicer program using the CT dataset, and arrows were drawn in Figure 1b to indicate the fractured area in detail. Table 1 summarizes the patient characteristics of the cohort.

The collected data were divided into training, validation, and test sets for the learning and evaluation of the deep learning models. Table 2 lists the configurations of the dataset used to train, validate and test the classification networks.

The window setting of each facial CT image was designated as hard tissue and bone window (window width: 2800; window level: 600), which has been reported in clinical trials to improve accuracy in determining bone fractures [18]. In addition, the collected cases were reconstructed with a 20 mm × 20 mm × 20 mm ROI dataset, including in all images the entire nasal bone, for learning based on 3D-CNN in a single network. After the radiologists manually choose the first slice including part of the nasal bone on the system’s graphical user interface (GUI), the algorithm automatically generated 3D cubic voxel data of the predetermined size starting with the slice chosen by the radiologists. The GUI is shown in Figure 2a.

The collected nasal ROI images were reprocessed into 3D data from each side of the isotropic voxel (64 px × 64 px × 64 px) and used in the deep learning experiments. An example of image pre-processing by the GUI is shown in Figure 2b. The range of pixel spacing of the obtained data was 0.717–1.071 mm, the mean was 0.837 mm, and its standard deviation was 0.069 mm. From this, the pixel spacing of all images was unified using the mean value (0.837 mm).

### 2.3. Computational Environment

The deep learning algorithms were implemented in Python 3.6.10 using the Keras 2.3.1 framework on a workstation with four NVIDIA RTX 2080Ti GPUs and 128 GB of RAM under the Ubuntu 14.04 operating system.

### 2.4. 3D CNN-Based Deep Learning Method

3D-CNN-based deep learning performs automatic feature extraction and selection and involves spatial dimensions in training the model to optimize performance [19]. Furthermore, in 3D deep learning, feature extraction and analysis are highly dependent on the model used for learning, and the high-level image features selected can vary depending on the depth of the model used. Therefore, 3D-CNN makes it possible to utilize the morphological and spatial information obtained from the various fracture forms of each patient’s medical images for learning.

Input data were obtained by cropping the nasal bone region from the facial CT images, and the entire nasal bone image was implemented in a three-dimensional isotropic voxel form. The residual neural network (ResNet) used in the experiment is a learning model with structures that minimize each learning step through skip connections, as shown in Figure 3. For that reason, it was selected to learn a single network with different depths and conducted 5-fold cross-validation to evaluate the performance of the network and validate the possibility of automatic nasal fracture diagnosis.

### 2.5. Performance Assessment

The evaluation of the performance of each model was performed using 549 test samples not used for learning. For performance evaluation, the results of each classifier were organized in a confusion matrix including the number of true-positive (*TP*), false-positive (*FP*), true-negative (*TN*), and false-negative (*FN*) predictions. In this variable, the positive cases mean a normal group, while the negative cases mean a fractured group. The sensitivity, specificity, and accuracy were calculated according to Equations (1)–(3). Equation (1) is formulated for sensitivity, refers to the test’s ability to correctly detect normal patients who do not have nasal fractures. Equation (2) means specificity, which refers to the test’s ability to correctly detect fractured patients on nasal bone. Lastly, Equation (3) refers to accuracy which means the test’s ability to correctly detect the patients who are classified according to each label within the entire data. The receiver operating characteristic (ROC) curves were built based on sensitivity and specificity. The performance of each model was evaluated based on the area under the ROC curve (AUC).
(1)$$Sensitivity=\frac{TP}{TP+FN}$$
(2)$$Specificity=\frac{TN}{TN+FP}$$
(3)$$Accuracy=\frac{TP+TN}{TP+FP+TN+FN}$$

## 3. Results

The binary 3D classification of nasal bone based on the facial CT using a single network with the 3D cubic voxel data approach achieved high performance. Table 3 shows the average 5-fold cross-validation results of each model in detail, representing the high performance in the automatic diagnosis of nasal fractures utilizing both 3D-ResNet34 and 3D-ResNet50. Table 3 reported that 3D-ResNet34 achieved 93.4% of AUC, 86.4% of sensitivity, 86.8% of specificity, and 86.2% of accuracy. 3D-ResNet50 achieved 94.5% of AUC, 87.5% of sensitivity, 87.8% of specificity, and 87.6% of accuracy as shown in Table 3 with higher performance compared to 3D-ResNet34. In results, it demonstrated the classification network with 3D-ResNet50 has higher performance in computer-aided diagnosing of fractured nasal bone compared to 3D-ResNet34. For Table 4, the average performance values of two deep learning models used in the experiments are presented. In this study, 94.0% of the average AUC, 87.0% of the average sensitivity, 87.3% of the average specificity, and 86.9% of the average accuracy of the deep learning models, suggesting the possibility of artificial intelligence (AI)-based nasal fracture diagnosis.

Figure 4 shows the ROC curves charting the performance of the classification learning systems. The difference in AUC values between the 3D-ResNet50 and 3D-ResNet34 was statistically significant (*p* < 0.01). Based on the ROC curves characterizing the performance of the classification systems, the classification network of 3D-ResNet50, which has a relatively higher depth of model learning, appeared to be more suitable to the data utilized in this experiment.

## 4. Discussion

Recently, AI-based CAD studies using medical imaging data have been reported as high performance, and many researchers have applied deep learning to improve diagnostic procedures based on a limited number of medical imaging data and diagnostic environments [20,21,22]. For example, in the case of diagnosing the fractures, the study of the deep learning-based classification of fractured regions was implemented for many areas of the human body (e.g., hips, humerus, ribs) and published [23,24,25]. In particular, deep learning models based on 3D-CNN are significant for CAD systems that utilize images to diagnose in medical fields around the world when trying to understand the three-dimensional human body [26,27]. However, the development of an AI-based automatic system using 3D voxel data extracted from the original CT images is less than that of studies using 2D input images. Therefore, this study aimed to verify performance by a 3D single network reported high performance likewise without a secondary network such as segmentation or rendering. It means that is more adequate in limited medical data, so research on this is essential.

This study aimed to aid diagnosis and verify the possibility of classifying fractured nasal bones using a 3D single-network deep learning model applied to the extracted facial CT images. It was proceeded by specifying the deep learning networks of 3D-CNN-based ResNet34 and ResNet50 and the cohort was used for learning. By comparing the performance of two models, this experiment makes us identify a more suitable classification model in the limited data given.

Compared to the performance of deep learning models including 3D-ResNet34 and 3D-ResNet50, in terms of average AUC, the classifier with 3D-ResNet50 was about 0.011 higher than with 3D-ResNet34. Although it’s a small difference, the result involved that in the case of 3D-ResNet34, it missed the meaningful high-level features of 3D voxel data through the lower layer, and only functions that are relatively less important and extracted from each lower layer degrade performance for the quality of the data.

In the analysis in detail when errors occur about 150 cases on average, both models reported errors in several cases in which the cubic voxel data included only minor fractures without depressed fractures or deviated nose. Figure 5a shows the examples of errors in the case of a normal nasal bone being misdiagnosed into a fractured nasal bone, and Figure 5b shows the case of a fractured nasal bone being misdiagnosed into a normal. As Figure 5, when there are minor fractures that are difficult to identify even in the image itself, they were confused with the shape of various noses and classified as normal. Also, when the shape of fracture is not far out of the normal nose shape categories, errors even occurred. In detail, like Figure 5a, if the space between nasal bone and nasomaxillary suture is relatively large due to natural deformation anatomically, the cases were classified as a fracture. On the contrary, like Figure 5b, if the fractured bone remained densely and the degree of deformation due to the trauma did not exceed a certain range, it was classified as normal. In shorts, the limitation of this study is that not include various deformation for learning. The minor fracture similar to normal such as greenstick and unilateral fracture result errors. Nevertheless, the overall performance of both models was similarly high to that reported for previous 3D classification of medical images for other fractures using multiple networks in recent decades. In addition, using facial CT images for automatic classification of the nasal fractured bone on a single network is novel. Although the study utilizes limited 3D nasal bone data, it was statistically significant and proved the possibility of CAD in diagnosing nasal fractures using 3D input data including the whole nasal region.

Therefore, a system able to perform a more sophisticated diagnosis of various fractures with relatively higher performance by future extensions of this model. Also, the future model can consider various fracture forms instead of simple binary classification, while increasing the learning cohort. More generally, data processing and model performance verification, such as those implemented in this work, are useful in the development of computer-aided systems utilizing a 3D-image field of view (3D-FOV).

## 5. Conclusions

For this paper, the study verified the possibility of an automatic diagnosis of nasal bone fractures using AI classification with 3D-CNNs in a facial 3D CT image voxel dataset. The high performance for classification was obtained for the automatic diagnosis of the presence or absence of nasal fractures with a single network. It reported a faster process with fewer weights than multiple networks, and as in other studies about classification for fracture, it showed high performance. Further learning and improvement could lead to future CAD models contributing to higher accuracy and reliability in the automatic classification of nasal fractures with further collected various shapes of the nasal bone. In addition, the extension of our system to an automatic fracture diagnosis system using 3D voxel data including the whole anatomic structure is expected to help increase the scope of AI-based human fracture diagnosis.

## Figures and Tables

**Figure 1 sensors-22-00506-f001:**
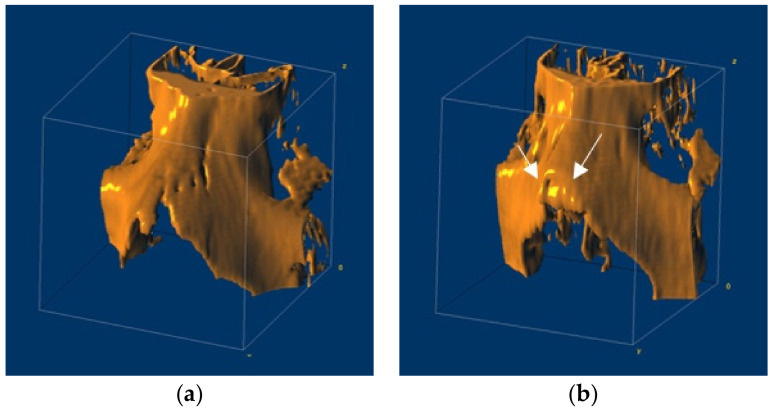
The 3D rendering reconstruction of nasal bones: (**a**) The reconstructed model of normal nasal bone; (**b**) The reconstructed model of the fractured nasal bone.

**Figure 2 sensors-22-00506-f002:**
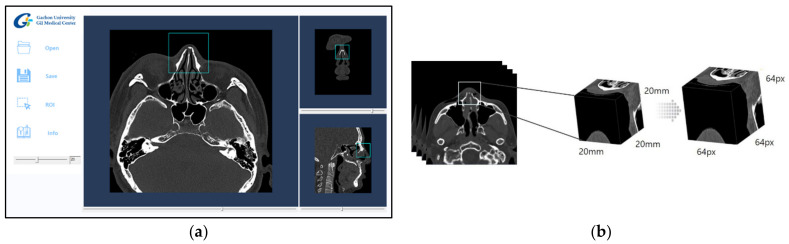
The preprocessing for constructing learning data.: (**a**) A graphical user interface of extracting cubic voxel data including overall nasal bone; (**b**) The process of data resampling.

**Figure 3 sensors-22-00506-f003:**
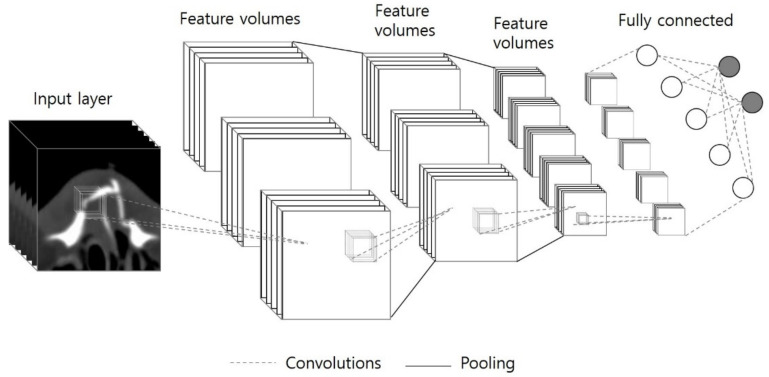
The architecture of 3D-ResNet.

**Figure 4 sensors-22-00506-f004:**
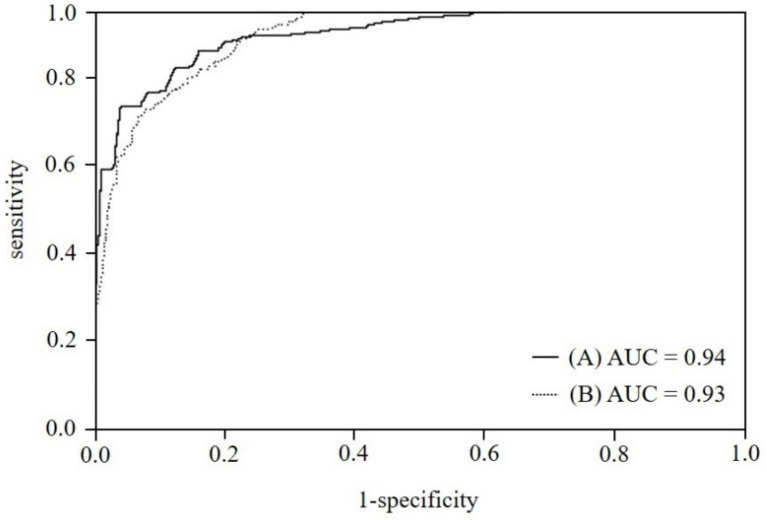
The ROC comparison of results; (**A**) The curve of 3D-ResNet50, (**B**) The curve of 3D-ResNet34.

**Figure 5 sensors-22-00506-f005:**
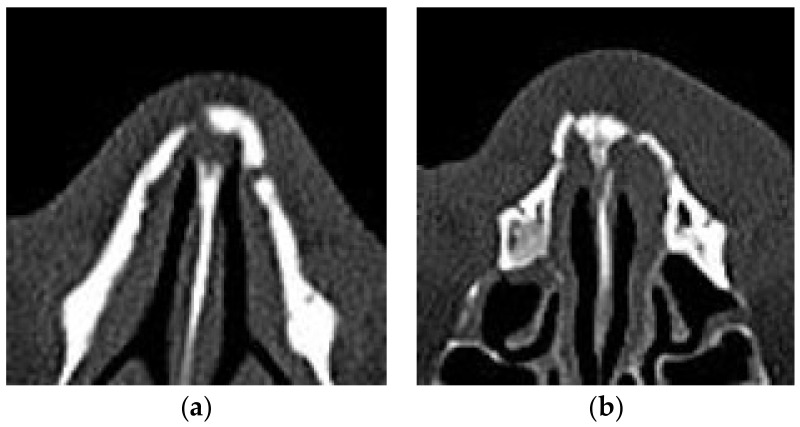
The example of misclassified errors: (**a**) the case of a normal nasal bone being misdiagnosed into a fractured nasal bone; (**b**) the case of a fractured nasal bone being misdiagnosed into a normal.

**Table 1 sensors-22-00506-t001:** Patient characteristics in the normal and fracture groups.

Characteristic	Summary	
	Normal Group	Fracture Group
Patients	N = 1350	N = 1185
Age, years (mean ± SD)	45.4 ± 20.7	46.5 ± 18.4
Sex	Male, 715; Female, 635	Male, 642; Female, 543

**Table 2 sensors-22-00506-t002:** Composition of deep learning dataset for training, validation, and test.

	Dataset	
	Normal	Fracture
Training	864	758
Validation	216	190
Test	270	237

**Table 3 sensors-22-00506-t003:** Comparison of AUC, sensitivity, specificity, and accuracy of the of learning models (3D-ResNet); (AUC, the area under the ROC curve; CI, confidence interval; ResNet, residual neural network.).

	AUC (95% CI)	Sensitivity (95% CI)	Specificity (95% CI)	Accuracy (95% CI)
3D-ResNet34	0.934 (0.927–0.941)	0.864 (0.862–0.867)	0.868 (0.866–0.872)	0.862 (0.861–0.863)
3D-ResNet50	0.945 (0.940–0.950)	0.875 (0.866–0.884)	0.878 (0.869–0.888)	0.876 (0.869–0.883)

**Table 4 sensors-22-00506-t004:** The Average of ResNet model results.; (Values are reported as mean ± SD. AUC, area under the ROC curve; ResNet, residual neural network; SD, standard deviation.).

	Classification Based on 3D-ResNets (*p* < 0.01)
AUC	0.940 (±0.01)
Sensitivity	0.870 (±0.01)
Specificity	0.873 (±0.01)
Accuracy	0.869 (±0.01)

## Data Availability

The dataset used and analyzed in this study are available from the corresponding author upon reasonable request.

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
