# Peer review of "A Study on 3D Deep Learning-Based Automatic Diagnosis of Nasal Fractures"

_sensors, 2022, doi:10.3390/s22020506_

Round 1

Reviewer 1 Report

Dear Authors

The paper is interesting. The paper organization is fine, the language requires significant revision. From my point of view, it would not have the potential to be considered in a publication at the current format.

Essential comment:

  • Language must be reviewed;
  • The methodology must be better presented, so other researchers can repeat your approach on their own examples;
  • It would be great if more examples or results added to the paper.

Other comments:

  1. It is highly recommended not to use the active tense beginning with “WE” in the academic manuscript. Therefore, please correct this issue in the revised version and use the passive tense.
  2. It is better to include a table of nomenclature in this paper since there is a plenty of acronyms, variables and abbreviations.
  3. What is the innovative point of this work? I mean advantages compared to the already published works? It is hard to find it along the paper.
  4. The figure 1 presents a 3D shape of the nasal bone, correct? How did you obtain it? I mean the image processing approach? Through point clouds? Please discuss this matter.
  5. Moreover, how did you reconstruct it? Through image processing tool? How?
  6. How did you formulate the equations 1 - 3?
  7. Conclusions must be rewritten. It must point out the main outcome of the work.

Very Best

The Reviewer

Author Response

Thank you.

Reviewer 2 Report

This manuscript is very well written, with minor writing errors in the discussion. Below are my comments

Introduction

The introduction is well written however, I would recommend that the authors remove the results of their study from the introduction and simply focus on the objectives of the study. 

Methodology

  1. Please provide IRB approval number in the methodology
  2. If available please provide mean information about time after fracture when the CT scan was conducted (i.e. mean time after fracture before scan = 1.2 days).
  3. Parts of the manuscript leave lots of assumptions. I would suggest further clarifying how data were divided into training, validation and testing
  4. Sections 2.3 and 2.4 have the same sub-titles. I would recommend changing 2.4
  5. The figures are greatly appreciated

Results

The results are very well written and clear

Discussion

You do a good job with the discussion however, there is significantly more information need, especially when you make statements such as "many studies". Please cite whenever you talk about studies that have been conducted in the past. There are several instances of this.

Additionally, there are several instances of grammatical error in your discussion section. While the rest of your manuscript has impeccable English, the discussion section has significant errors.

Can you elaborate more on why the normal bone may have been classified as a fracture?

What are the limitations of the study?

Author Response

Thank you.

Round 2

Reviewer 1 Report

Dear Authors

Good improvement!

Can be accepted at this stage.

Regards

The Reviewer 

Author Response

Thank you.

Reviewer 2 Report

Thank you for taking the time to address my comments. Besides a few minor grammatical errors you did a great job!

Author Response

Thank you.
